# Plasmon-Induced Transparency for Tunable Atom Trapping in a Chiral Metamaterial Structure

**DOI:** 10.3390/nano12030516

**Published:** 2022-02-01

**Authors:** Zhao Chen, Yaolun Yu, Yilin Wang, Zhiling Hou, Li Yu

**Affiliations:** 1College of Mathematics and Physics, Beijing University of Chemical Technology, Beijing 100029, China; wylwyl@mai.buct.edu.cn (Y.W.); houzl@mail.buct.edu.cn (Z.H.); 2State Key Laboratory of Quantum Optics and Quantum Optics Devices, Shanxi University, Taiyuan 030006, China; 3Nanophotonics and Optoelectronics Research Center, Qian Xuesen Laboratory of Space Technology, China Academy of Space Technology, Beijing 100094, China; yuyaolun@qxslab.cn; 4School of Science, Beijing University of Posts and Telecommunications, Beijing 100876, China

**Keywords:** plasmon-induced transparency, metamaterials, high refractive-index dielectric, atom trapping

## Abstract

Plasmon-induced transparency (PIT), usually observed in plasmonic metamaterial structure, remains an attractive topic for research due to its unique optical properties. However, there is almost no research on using the interaction of plasmonic metamaterial and high refractive index dielectric to realize PIT. Here, we report a novel nanophotonics system that makes it possible to realize PIT based on guided-mode resonance and numerically demonstrate its transmission and reflection characteristics by finite element method simulations. The system is composed of a high refractive-index dielectric material and a two-dimensional metallic photonic crystal with 4-fold asymmetric holes. The interaction mechanism of the proposed structure is analyzed by the coupled-mode theory, and the effects of the parameters on PIT are investigated in detail. In addition, we first consider this PIT phenomenon of such fields on atom trapping (^87^Rb), and the results show that a stable 3D atom trapping with a tunable range of position of about ~17 nm is achieved. Our work provides a novel, efficient way to realize PIT, and it further broadens the application of plasmonic metamaterial systems.

## 1. Introduction

Electromagnetically induced transparency (EIT) is an electromagnetic phenomenon in three-level atomic systems owing to the interference between excitation pathways and the atomic upper level [1,2]. The EIT profile possesses a distinctly narrow window, which remains valid for sensing, switches, nonlinear and slow-light devices [3,4,5]. Recent studies have shown that the EIT effect can be achieved in nonquantum photonics systems, such as metamaterials [6,7,8], photonic crystals [9,10,11], and metal–insulator–metal (MIM) waveguides [12,13,14]. As mentioned, metamaterials that can be designed and manipulated to obtain extraordinary properties have attracted considerable attention [15,16,17]. In general, metamaterials can be divided into plasmonic [18] and all-dielectric metamaterials [19]. Plasmonic metamaterials, whose microstructure units are composed of noble metals (e.g., Au, Ag), have surface plasmon-related properties [6,7,8]. Here, the EIT is usually called plasmon-induced transparency (PIT), while all-dielectric systems generally use high refractive indices and low-loss materials, such as Si, to design the microstructure units [20,21,22]. Metamaterials-based EIT effect has essential applications in optical storage, nano-sensing, filtering, and slow light [6,7,8,20,21,22]. Yang et al. proposed a high Q-factor (Q ≈ 483) cavity based on the all-dielectric EIT effect [20]. Lior et al. achieved a group delay of about ~8 fs based on the PIT effect [23]. However, there are few reports on studying the EIT phenomenon combined with plasmon systems and high refractive index materials. 

In this paper, we combine the field enhancement advantages of the plasmonic structure and the low damping losses characteristics of the high refractive index dielectric materials to achieve the PIT effect. This hybrid system is analyzed by temporal coupled-mode theory (CMT) and confirmed by finite element method (FEM) simulations. Simulation results show that a transparency window appears in the reflection spectrum, tuned by the structure parameters. Furthermore, the atom-trapping characteristic of this system is also investigated, and only about 0.63 mW input optical power and 1 mK trapping depth can be achieved for neutral atoms. By changing the circularly polarized light phase, the trapped atom can move back and forth up to 17 nm. This kind of hybrid system is easy to be fabricated and integrated, and the proposed method of achieving PIT provides a promising possibility for functional optical components in highly integrated optics.

## 2. Structures and Simulation Results

The schematic diagram of our structure with relevant parameter symbols is proposed in Figure 1a, which consists of a 4-fold asymmetric Ag holes [24], silica (*n*_SiO2_ = 1.45), and silicon [25]. The incident light is set to circular polarization light (CPL), and the incident direction is from +z to −z. In order to obtain a qualitative understanding of the optical waveguide, we analyze the transmission/reflection properties of the hybrid system by CMT [26,27,28], as shown in Figure 1b. Here, *a* (Ag holes) or *b* (High refractive index materials, Si) are the resonance amplitudes inside the resonator, which are normalized such that |*a*|^2^ or |*b*|^2^ corresponds to the energy in the resonator. *S_i+_* and *S_i−_* (*i* = 1, 2) denote the waveguide mode amplitudes of the incoming and outgoing waves into the resonator. *S* is normalized such that |*S*|^2^ represents the power of the wave [27]. The time evolution of the amplitudes of the cavity in a steady state can be described as [26,27](1)$$\frac{da}{dt}=j\omega_{a}a-(\frac{1}{\tau_{ia}}+\frac{1}{\tau_{a}})a+\sqrt{\frac{1}{\tau_{a}}}S_{1+}-j\mu b$$(2)$$\frac{db}{dt}=j\omega_{b}b-(\frac{1}{\tau_{ib}}+\frac{1}{\tau_{b}})b-j\mu a$$Here, the *ω_a_* (*ω_b_*) and 1/*τ_ia_* (1/*τ_ib_*) are the resonance frequency and the intrinsic cavity loss rate of the cavity *a* (*b*), respectively. 1/*τ_a_* and 1/*τ_b_* are defined as the decay rates of the cavity amplitude *a* and *b* into port 1 and port 2, respectively. The outgoing waves in the resonant waveguide should satisfy a steady-state relation:(3)$$S_{1+}=\sqrt{\frac{1}{\tau_{a}}}a,\;S_{2-}=\sqrt{\frac{1}{\tau_{b}}}b$$

We choose port 1 as the input port and obtain *S*_2+_ zero. Therefore, the transfer function of the system can be derived as
$$r=\frac{S_{1-}}{S_{1+}}=\frac{[j(\omega -\omega_{b})-\frac{1}{\tau_{ib}}-\frac{1}{\tau_{b}}]\frac{1}{\tau_{a}}}{[j(\omega -\omega_{a})+\frac{1}{\tau_{ia}}+\frac{1}{\tau_{a}}][j(\omega -\omega_{b})+\frac{1}{\tau_{ib}}+\frac{1}{\tau_{b}}]+\mu^{2}}\;\mathrm{and}$$
(4)$$t=\frac{S_{2-}}{S_{1+}}=\frac{-j\mu \sqrt{\frac{1}{\tau_{a}\tau_{b}}}}{[j(\omega -\omega_{a})+\frac{1}{\tau_{ia}}+\frac{1}{\tau_{a}}][j(\omega -\omega_{b})+\frac{1}{\tau_{ib}}+\frac{1}{\tau_{b}}]+\mu^{2}}$$

Additionally, the reflection and transmission spectrum of the hybrid system are
(5)$$R=|r{|}^{2}\;\mathrm{and}\;T=|t{|}^{2}$$

It can be seen from Equation (4) that the hybrid system is a coupling of two simple systems, which is consistent with the results in ref. [28].

There are two paths for incident light waves, as shown in Figure 1c. We assume that the ground state of the light wave is |0>, and the excited state of the cavity *a* (*b*) is |1> (|2>). There are two different transition pathways |0>→|1> and |0>→|1>→|2>→|1> with which an atom in the ground state can experience a transition to the excited state. The two paths interfere with each other, creating an EIT-like phenomenon in the reflection spectrum [1].

To verify the above analysis, we calculate the optical response (Transmission (T) and Reflection (R) spectrum) of the proposed system, shown in Figure 1a, by COMSOL Multiphysics 5.2a (Stockholm, Sweden) based on FEM (see details in Appendix A). The optimized parameters are set as follows: *P* = 680 nm, *R* = 150 nm, *r* = 50 nm, *ϕ* = π/2, *h* = 215 nm, *t* = 215 nm, and *d* = 215 nm. The calculated results are shown in Figure 1d, and it can be found that only the Lorentzian curve appears when there are only Ag holes (Figure 1d upper maps). In contrast, when Si material is added, the reflection spectrum of the system shows a PIT-like phenomenon (Figure 1d lower maps). The original peak in the transmission spectrum becomes a valley, and the electromagnetic-induced absorption (EIA) phenomenon appears. These results are consistent with the above analysis. Figure 2a,b display the modules of electric |E| and magnetic field |H| distributions at λ = 748 nm, 755 nm, and 760 nm, respectively. It can be seen that when λ = 748 nm and λ = 760 nm, the surface plasmon mode is excited, and the energy of the light field is mainly concentrated on the Ag holes, showing the transmission peaks (reflection valleys). At λ = 755 nm, most of the energy is reflected, showing a transmitted valley (reflection peak). Hence, we have achieved an EIT-like phenomenon in the reflection spectrum by using Ag holes and high refractive index materials without increasing the total size of the system, which has rarely been reported before in this way and is of great significance in device miniaturization and all-optical integration. Here, since the chiral response characteristics of this system are not obvious, there is almost no difference between the transmission and reflection spectrum under the excitation of RCP and LCP (see details in Appendix B). The distinction between RCP and LCP is no longer made, unless otherwise stated.

## 3. Optical Properties of the System with Different Parameters

Successively, we investigated the influence of parameter variations on the optical properties to verify the spectral tunability of the system. Only one parameter is changed during the calculation, while the other parameters are set the same as in Figure 1a and Figure 3a,c show the corresponding calculation results for *P*, *d*, and *ϕ* variation, respectively. It can be seen that with the increase in period *P* or silicon thickness *d*, the position of the transparent window presents a linear redshift with ΔP/Δλ ≈ 10 nm/3 nm or Δd/Δλ ≈ 5 nm/9 nm. With the increase in radian *ϕ*, the transparent window displays a nonlinear redshift, and when *ϕ* is much larger than π/2, the transparent window becomes very irregular due to the parameter mismatching. Further, we calculated the sensing characteristics of the system, i.e., changing the refractive index of intermediate material silica n_SiO2_, and the results are shown in Figure 3d. It can be found that the position of the transparent window is linearly redshifted with the increase in the refractive index (Δn_SiO2_/Δλ ≈ 0.05/6 nm), so the sensitivity of the refractive index sensor obtained can reach 120 nm/RIU [29]. In addition, this kind of PIT feature can also be applied for filter [30], nanophotonic biosensors [31], and ultrafast all-optical switching [32]. The above results illustrate that the system has good tuning characteristics. In particular, introducing the high refractive index materials would increase the system’s tunability and, further, the structure’s practicability.

## 4. Trapping Characteristics

Neutral atom trapping plays a vital role in quantum computing and data storage. The minimum value of the local electric field generated by the near-field scattering effect of periodic surface plasmons is the best region to realize the atom trapping with detuned blue light [33,34,35]. Herein, we first unite the PIT phenomenon with atom trapping through the structure proposed in Figure 1a and discuss in detail the PIT effect on the trapping characteristics of neutral atoms. In particular, we investigate the spatial distribution of trapping potential at the incident wavelengths of λ = 748 nm, 755 nm, and 760 nm (shown in Figure 1d), and the normalized electric field modulus distributions (x-z plane, y = 0 nm) are shown in the inset in Figure 4a. A local electric field minimum appears in the plot, suggesting the blue-detuned optical trap for neutral atoms [36]. Here, *ds* represents the distance between the location of the minimum local electric field (trap center) and the structure’s surface. We take ^87^Rb atoms, whose primary transition wavelengths are 794 nm and 780 nm, as an example to illustrate the trapping characteristics of the proposed system. 

A neutral atom in an electric field *E* experiences repulsive optical dipole potential, which is given by [36]
(6)$$U_{opt}=-\frac{1}{4}\alpha |E{|}^{2}$$
where α is reduced polarizability and can be derived from reference [37]. Correspondingly, the polarizability is α = −5.08 × 10^−38^ [F∙m^2^], −6.44 × 10^−38^ [F∙m^2^], −7.87 × 10^−38^ [F∙m^2^] for the three incident wavelengths λ = 748 nm, 755 nm, and 760 nm, respectively. Additionally, surface potential plays an important role in the trapping characteristics when the atom is very close to the surface (within 100 nm) [38,39]. However, in our case, *ds* is about 300 nm away from the surface, while the surface potential can be neglected in the calculation, which can be attributed to the near-field scattering properties of SPPs [33,34,35]. Therefore, the total potential depends on the optical potential *U_opt_* only, which can be easily tuned through |E| by optimizing the structural parameters. At this point, the location of the minimum local electric field is also the trap center of the neutral atom. Figure 4a shows the *U_opt_* distributions along the green dashed line obtained from Eqaution (6) under different incident wavelengths with input power P_0_ = 1.0 mW. The arrow diagram shows the enlarged trapping positions. It can be found that ds = 367 nm (RCP) or 350 nm (LCP) at λ = 748 nm, ds = 364 nm (RCP) or 350 nm (LCP) at λ = 755 nm, and ds = 364 nm (RCP) or 349 nm (LCP) at λ = 760 nm, respectively. Such long trapping distances further ensure the stability of the system. In general, the effective dipole trap depth U_eff_ = 1 mK is commonly used in single-atom trapping experiments [36]. In our case, to achieve stable atom trapping, the incident light power P_0_ required is about 1.0 mW (λ = 748 nm), 10 mW (λ = 755 nm), and 0.63 mW (λ = 760 nm), respectively. Ultra-low optical power can effectively reduce the photon scattering rate and increase the trapping lifetime of the trapped atoms [34]. Therefore, λ = 748 nm and λ = 760 nm can be regarded as the most appropriate trapping wavelengths. In addition, our system can achieve a one-dimensional atomic array of captivity, which has essential applications in resonance fluorescence and manipulation of cold atom arrays. A stable three-dimensional atom array trapping independent of the surface potential with ultralow optical power is demonstrated (see the trapping potential distributions of z = ds-plane details in Appendix C).

Although the chiral response of the system is not apparent, for the trapped atoms, this slight difference can be used to tune their position. Figure 4b quantitatively displays the position of the trapped atom at any phase difference *ΔΦ* (see Appendix A) within one phase circle (0 2π) at λ = 748 nm and λ = 760 nm. Clearly, during the transformation process of left and right optical rotation, the moving range of the trapped atom is about 17 nm and 15 nm for λ = 748 nm and λ = 760 nm, respectively. Here, we can detect the strength of the structural chiral optical response by the size of the movement range of the trapped atoms. Due to the weak optical chirality response of the system, the regulation range of trapped neutral atoms is only 15–17 nm. However, it is very difficult to balance both PIT and atom trapping in one structure because the trapping wavelength range of the atom is defined. Our work provides a reference for the realization of micro–nano devices with more complex functions in the future. In addition, the proposed system can trap all neutral atoms whose primary transition wavelength is longer than 760 nm, such as ^133^Cs, for which the practicability of the trapping system dramatically increases. Henceforth, we have achieved stable neutral atom trapping through the proposed structure, which has significant applications in preparing atomic gyroscopes and atomic clocks. In particular, large arrays of individually controlled trapped atoms are a very promising platform for quantum engineering applications [40,41].

In the future study of PIT, metamaterials with more significant chiral responses can be designed to control the presence or absence of PIT windows in the transmission spectrum to achieve a more extensive tuning range for neutral atoms based on the PIT effect simultaneously. This kind of nanostructure can be easily fabricated by the modern nano etching technology, e.g., templated plasmonic substrates [42] and electron beam lithography [43]. The micro–nano structure design, integrating multiple functions, is a trend of the development of optical devices in the future.

## 5. Conclusions

In conclusion, we have demonstrated a stable 3D atom trapping based on PIT effect in a chiral metamaterial system consisting of a 4-fold asymmetric Ag holes and high reflective index material Si. The interference of light waves in different paths produces a PIT window in the reflection spectrum, and its tunability is simulated in detail by FEM. Due to the near-field scattering effect of SPPs, the proposed system yields a good trapping characteristic for neutral atoms, e.g., ^87^Rb, and the position of the trapped atoms can be easily tuned by altering the phase difference of the incident light. Our design provides a new idea for the realization of PIT, and it also broadens the application of PIT in atom trapping, which is of great significance in slow light, resonant fluorescence, and optical sensing.

## Figures and Tables

**Figure 1 nanomaterials-12-00516-f001:**
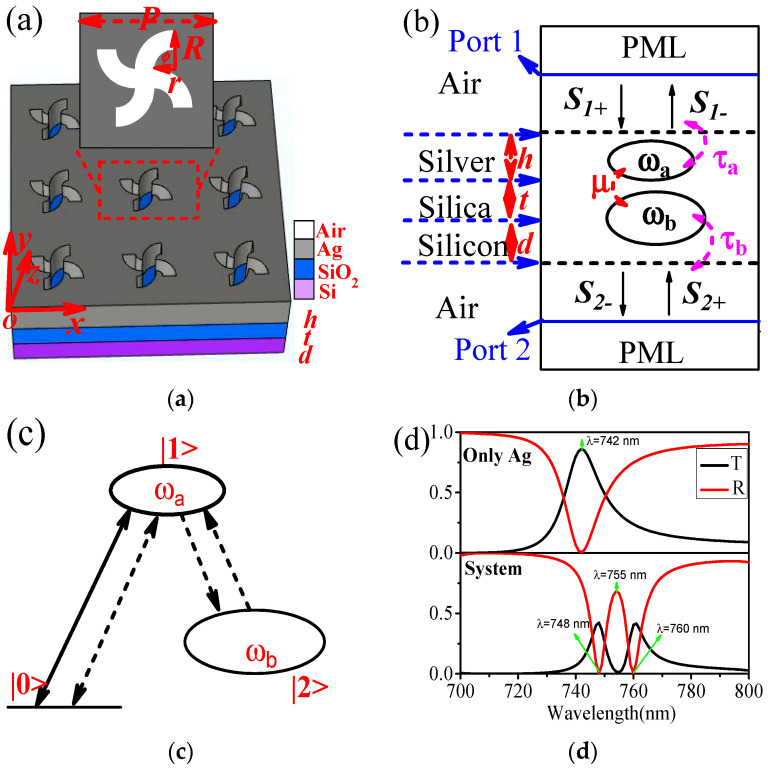
(**a**) Schematic of chiral metamaterials. The red dashed line shows the unit cell of the presented arrays and the main geometrical parameter symbols. (**b**) CMT parameter description diagram. (**c**) EIT three−level system. (**d**) Transmission and reflection spectrum of the only Ag structure (upper) and whole (lower) system. Here, P, R, and r display the period, outer radius, and inner radius of the unit cell. *ϕ* is the arc angle of the 4−fold asymmetric Ag holes; h, t, and d represent the thickness of Ag, SiO_2_, and Si, respectively.

**Figure 2 nanomaterials-12-00516-f002:**
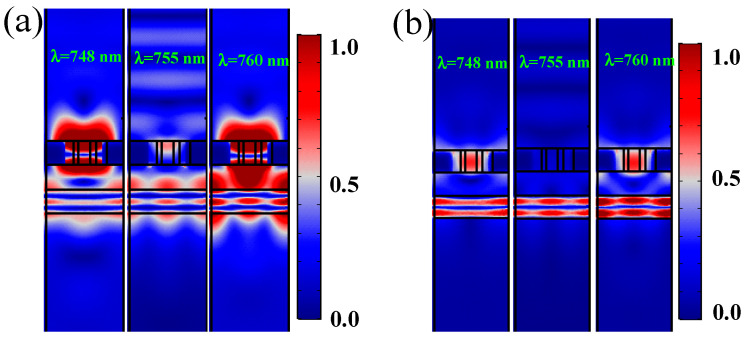
Normalized field distributions of (**a**) |E| and (**b**) |H| at λ = 748 nm, 755 nm, and 760 nm. Here, *P* = 680 nm, *R* = 150 nm, *r* = 50 nm, *ϕ* = π/2, *h* = 215 nm, *t* = 215 nm, and *d* = 215 nm.

**Figure 3 nanomaterials-12-00516-f003:**
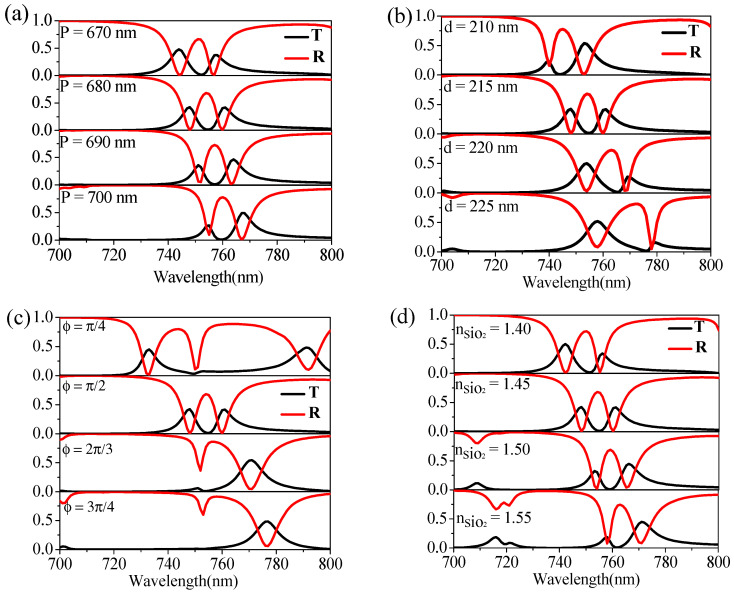
Influence of parameter variations of (**a**) period *P*, (**b**) thickness of silicon *d*, (**c**) arc angle *ϕ,* and (**d**) refractive index of silica *n_SiO2_* on transmission and reflection spectra.

**Figure 4 nanomaterials-12-00516-f004:**
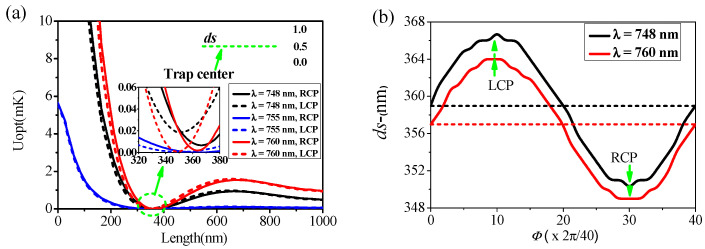
(**a**) Distributions of *U_opt_* along the green dashed line at λ = 748 nm, 755 nm, and 760 nm for RCP and LCP incidence. (**b**) The variation law of *ds* with change in phase difference *ΔΦ* at λ = 748 nm and λ = 760 nm. Inset shows the normalized field distributions at the trapping wavelength.

## Data Availability

Not applicable.

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
