# Peer review of "Plasmon-Induced Transparency for Tunable Atom Trapping in a Chiral Metamaterial Structure"

_nanomaterials, 2022, doi:10.3390/nano12030516_

Round 1
Reviewer 1 Report
The paper presents original research (modeling) of a specific structure composed of silica, silicon, and silver. Even so, the results are not of general soundness and global significance, they should be interesting to a wider group of readers due to the well-presented idea on the structure design.
I would recommend publishing the paper as is. The presentation and language are very good, however, some minor optional editorial could be recommended to make everything perfect.
Author Response
Thanks for your suggestion.
Reviewer 2 Report
This is an interesting theoretical work on observing plasmon induced transparency in suitably designed layered metamaterial structures. An incident wave couples to two waveguide modes in the metamaterial structure, and their interference leads to a spike-like feature in the reflection spectrum. The same structure is also shown to be able to function as a stable 3D atom trap.
The presentation of the results and of the underlying theory is sufficiently clear, and the conclusions are indeed technically sound. The manuscript, however, could be somewhat further improved and become even more interesting for a general readership if the authors could also address the following points:
1) How 'sharp' can the central spike in the R spectrum become in the presence of realistic material optical losses? is there any room for further improvement by using e.g. atoms or gain? [see/cite, e.g., Nature Communications 5, article number: 3808 (2014); Nature Photonics 8, 685-694 (2014); Physica B: Condensed Matter 407, 4066-4069 (2012), etc]
2) How much is the attained delay in the propagation of the incident wave through the structure?
3) How much is the attained delay-bandwidth performance of this resonance-based scheme? can it approach or exceed unity?
4) Is the scheme (topologically) stable in the optogeometric-parameters space? that is, do small imperfections, roughnesses, etc adversely affect its functioning / performance?
5) What applications could benefit from the scheme? [for instance, is the scheme suitable for sensing applications? - see, e.g., Nature Nanotechnology 17, 5-16 (2022)]
Reviewer 3 Report
This research is a simulation-based study on plasmon-induced transparency of metamaterial nanostructures for atom trapping. I think that a research subject is suitable for publication in Nanomaterials-MDPI. On the other hand, I decided that this manuscript can be published only when the following contents are sufficiently supplemented.(1) I think that readers' understanding should be improved when the meaning and specification of each length and symbol are specified in the figure description of Figure 1.
(2) In Figure 1(d), a peak wavelength mark is required for the case of only Ag.
(3) The explanation of the part of '3. Optical properties of the system with different parameters' should be sufficiently supplemented. The meaning of each graph in Figure 3, especially the relationship between each parameter of the nanostructure and peak wavelengths, should be derived to indicate meaningful results.
(4) In a section of Conclusion, the authors briefly presented the meaning and application of the simulation-based research results, which should be supplemented in detail. In particular, studies to which the results of this study can be applied should be presented with reference papers.
(5) Contents on how to implement this structure experimentally should be added to a section of Discussion, and quantitative performance improvement should be presented compared to other nanometamaterials.
(6) I think that the height and width information of Figure 2(a) and (b) must be added to the figure to improve readers' understanding.
(7) The size and arrangement of each graph in the figure are not constant, so they should be aligned and shown well.
Reviewer 4 Report
The authors develop idea on Plasmon-induced transparency for tunable atom trapping in a chiral metamaterials structure, the paper is well written. They have demonstrated a stable 3D atom trapping based on PIT-effect in a chiral metamaterial system consisting of a 4-fold asymmetric Ag-holes and high reflective index material S. These designs provides a new idea for the realization of PIT, and it also broadens the application of PIT in atom trapping, which is of great significance in slow light, resonant fluorescence, and optical sensing.
Author Response
Thanks for your suggestion.
Round 2
Reviewer 3 Report
I confirmed that all points I commented on were properly supplemented and that this manuscript is ready to be published in Nanomaterials. One small thing is that Figs. or Eqs. in a main context should be processed in bold, which should be commented by the editorial department so that it can be corrected before publication.